# The Effect of Pyrolysis Temperature and the Source Biomass on the Properties of Biochar Produced for the Agronomical Applications as the Soil Conditioner

**DOI:** 10.3390/ma15248855

**Published:** 2022-12-12

**Authors:** Michal Kalina, Sarka Sovova, Jiri Svec, Monika Trudicova, Jan Hajzler, Leona Kubikova, Vojtech Enev

**Affiliations:** Faculty of Chemistry, Brno University of Technology, Purkynova 464/118, 61200 Brno, Czech Republic

**Keywords:** agriculture, biochar, biomass, pyrolysis, temperature, soil conditioner

## Abstract

Biochar is a versatile carbon-rich organic material originating from pyrolyzed biomass residues that possess the potential to stabilize organic carbon in the soil, improve soil fertility and water retention, and enhance plant growth. For the utilization of biochar as a soil conditioner, the mutual interconnection of the physicochemical properties of biochar with the production conditions used during the pyrolysis (temperature, heating rate, residence time) and the role of the origin of used biomass seem to be crucial. The aim of the research was focused on a comparison of the properties of biochar samples (originated from oat brans, mixed woodcut, corn residues and commercial compost) produced at different temperatures (400–700 °C) and different residence times (10 and 60 min). The results indicated similar structural features of produced biochar samples; nevertheless, the original biomass showed differences in physicochemical properties. The morphological and structural analysis showed well-developed aromatic porous structures for biochar samples originated from oat brans, mixed woodcut and corn residues. The higher pyrolysis temperature resulted in lower yields; however, it provided products with higher content of organic carbon and a more developed surface area. The lignocellulose biomass with higher contents of lignin is an attractive feedstock material for the production of biochar with potential agricultural applications.

## 1. Introduction

Biochar is a porous, carbonaceous material that is produced by the thermal decomposition of carbon-rich biomass materials in the absence of air. This ubiquitous black organic material attracts numerous scientific groups around the world due to its hyper-versatile utilization in agriculture, environmental engineering and basic chemical technologies [1,2]. The European Biochar Certification [3] defines as highly attractive those utilizations where biochar is applied in a way in which the contained carbon remains stored as a long-term carbon sink (e.g., utilization in agriculture as the soil conditioner) or replaces fossil carbon in industrial manufacturing (e.g., metallurgy [4]). The attractive chemical properties of biochar produced for agronomical utilization as a soil supplement are associated with a positive effect on a reduction in the total amount of greenhouse gasses released from the soil decreased leaching of macro- (P, N, Mg, Ca, K) and micronutrients (Al, Fe, Cu, Cr, Mn, Ni, Zn), plant health and growth enhancement, reduction in the mobility of various heavy metals (e.g., As, Cu, Pb, Cr, Cd) and positive effects on the soil microorganism communities [1,5]. The possible agronomical potential of biochar is also closely linked with its physicochemical properties, morphology and texture [6]. The idea of biochar utilization in agriculture has the historical consequence as the direct proof of the positive effect of biochar on soil properties is a Hortic anthrosol of the Amazon (*Terra preta*) [7]. This unusually fertile black soil contains a large amount of pyrogenic material (biochar) originating from ancient soil management practices, which persisted in the soil for centuries.

The crucial biochar physicochemical properties and structural characteristics are closely linked to the conditions used during the pyrolysis of biomass feedstocks (temperature of pyrolysis, heating rate, pressure, type of used carrier gas and its flow rate and residence time), the origin of the used source biomass material (the content of cellulose, hemicellulose or lignin), the residual biomass moisture and its particle (grain) size [8]. Generally, the temperatures used for the production of biochar samples vary between 200 and 700 °C [1]; for the production of biochar samples with potential agricultural applications, temperatures between 450 and 600 °C are generally accepted as the optimal [9]. High temperature applied to biomass materials transforms the organic matter in biomass into three different components—gas (e.g., methane and other hydrocarbons), liquids (formed by condensation of gases) and solids (charcoal or biochar). The high temperature induces the polymerization of molecules, which leads to the formation of relatively large structures (including both aromatic and aliphatic compounds), as well as the thermal decomposition of some components of the feedstock material into smaller molecules [1]. The increase in pyrolysis temperatures results in the production of biochar with increased persistence due to resistance to microbial and chemical decomposition in soil [10]. The residence time, according to the literature [8], also affects biochar yield and its internal porous structure due to the repolymerization of vaporized molecules, which can take place at higher residence times, resulting in a minor increase in biochar yield and a greater extent of the internal porous structure development. Moreover, the characteristics of biochar and its yield can also be affected by the selection of carrier gas (often used nitrogen, argon or water vapor), its flow rate and a selection of pyrolysis type [8,11].

The second crucial aspect affecting biochar production is the origin of a biomass feedstock material [2], which can vary from wood-based products (e.g., woodcut, woodchip, sawdust, wooden pellets and tree bark) to plant-based materials (e.g., wheat straw, leaves, husks, seeds and cobs) and organic wastes materials (e.g., manure and sludges) [12,13]. In total, the wide variety of potential biomass feedstock materials, together with the opportunity to utilize a huge extent of possible pyrolysis production conditions, enable the production of biochar samples with rather variable properties of produced material [14]. The wooden-based feedstock materials generate more resistant biochar with total carbon contents of up to 80%. On the other side, biomass with high lignin content (e.g., olive husk, woodcut, grass, vegetable) produces biochar with even higher carbon content. Therefore, at the same temperature, lignin loss of material due to pyrolysis is less than half of cellulose loss [13]. Besides organic carbon, hydrogen, oxygen, nitrogen, phosphorus and sulfur can also be identified in the structure of biochar, and their presence can significantly contribute to the heterogeneity of biochar surface and its reactivity [15]. During the pyrolysis process, volatile compounds of biomass such as functional groups containing O, H and S are volatilized, which leads to the accumulation of the non-volatile elements such as C and partially also N, P, K, Ca, Mg, Fe, etc. The stability is closely linked to the O/C ratio [12], which can vary in the range from 0.05 to 0.7. The content of other nutrients depends on the origin of the used feedstock biomass (lignin-based vs. cellulose-based). Biomass moisture content and feedstock particle (grain) size represent additional important parameters of biochar production as the biomass feedstock can contain either internal water vapor, chemically bound water in pores or free liquid water, which all in total can represent up to 60 wt.% of total mass [8]. Pyrolysis of biomass containing various types of internal moisture results in a decrease in heat transfer effectivity as the fraction of the heat energy is connected with moisture evaporation. For these reasons, the addition of the biomass drying step can contribute to a significant reduction in the necessary heat energy and also lower the necessary production time and production costs.

The literature provides a wide variety of published research dealing with the optimization of biochar application dosage and the corresponding agronomical impacts on soil and plant growth and yields [1,2,5,15,16]. Kocsis et al. [16] defined the application dose of biochar between 1 and 2.5 wt.% of dry soil (corresponding dosage between 20 and 50 t/ha of soil) as the optimal for the support of yields (22% yield increase in a model plant—*Zea Mays*). Moreover, these application doses have a positive effect on soil pH, cation and nutrition retention (mainly K; P; and partially Mg, Na and N) and soil microbial activity. These findings are supported by other published research [17,18,19], where the authors concluded regardless of the well-described positive impact of biochar on the horticultural and agricultural systems, the application doses above 20 t/ha are too expensive (considering the production costs). The production of biochar and its utilization as a soil conditioner is connected with the carbon neutrality assessment. Verheijen et al. [2] pointed out that the role of biochar is also connected with its effects on C cycling and mobility in soil. The global flux of C from soil to atmosphere in the form of CO_2_ is approximately 60 Gt of C per year, mainly as the result of microbial respiration (microbial decomposition of soil organic matter). Components of biochar introduced into the soil provide a considerably more stable form of organic structures compared to soil organic matter. This results in so-called carbon negativity of the process, as carbon inputs into the soil through the application of biochar as the stable C source are increased greatly compared to its output through soil microbial respiration. Lehman et al. [1,7] described the potential of a global C-sequestration of 0.16 Gt per year through the use of current biomass waste materials from forestry and agriculture and urban wastes for biochar production.

Despite the positive effects of biochar on soil properties and soil fertility, the biggest issue obstructing its broader agrochemical utilization is nowadays still hidden in the lack of description of possible routes how soil and soil organic matter properties can be affected by different types of biochar. Additionally, the mutual interconnection of the effect of biomass feedstock selection and the setting of optimal pyrolysis conditions on the properties of produced biochar is not fully described. Both these agronomically important aspects guiding the properties of produced biochar are addressed in the present manuscript. The main aim of our research is to focus on a description of the effect of source biomass origin (woody, non-woody, commercially composted biowaste) as well as the corresponding effect of pyrolysis conditions (temperature, residence time) on resulting biochar properties. The attention is paid mainly to the properties of biochar, which are significant for the agronomical utilization of this ubiquitous black carbon-rich material.

## 2. Materials and Methods

### 2.1. Materials

The biomass feedstocks for biochar production originated from oat brans (Mlyny Vozenilek, spol. s.r.o., Predmerice, Czech Republic), mixed woodcut (Central composting plant of Brno, SUEZ CZ, a.s., Brno, Czech Republic), corn residues (residual biomass from other research at Faculty of Chemistry, Brno University of Technology, Brno, Czech Republic) and commercial compost (Black Dragon, Central composting plant of Brno, SUEZ CZ, a.s., Czech Republic). All these biomasses belong to waste materials, which fulfill the recent action plans of the European Commission for the maximal use of natural resources without undesirable wastes.

### 2.2. Methods

#### 2.2.1. Pyrolysis of Biomass Feedstocks

The biomass samples (oat brans, mixed woodcut, corn residues and commercial compost) were air-dried at 45 °C (to reduce the residual biomass moisture), homogenized using an electric chopper (R-5115, Rohnson, Vassilias, Greece) and dry-sieved (size fraction below 2 mm). The pre-treated biomass samples were subjected to pyrolysis under a driven oxygen-free atmosphere using a laboratory furnace with vacuum and variable controlled atmosphere of pure gases CLASIC 501 (Classic CZ s.r.o., Revnice, Czech Republic) with integrated advanced PLC controller Vision 280 (Unitronics GmbH, Esslingen, Germany). Samples were evacuated in the furnace to clear off the oxygen, and pyrolysis was conducted under a nitrogen atmosphere (flow of 5 dm^3^/min) with a temperature gradient of 5 °C/min up to final temperatures of 400, 500, 600 and 700 °C. The furnace temperature is maintained according to the program by molybdenum disilicide heating elements controlled by calibrated thermocouple and the furnace computer. The inert nitrogen atmosphere flow assists with the heat distribution in the furnace. The target pyrolysis temperature was kept in the operating furnace for 10 and 60 min (residence time). After cooling still in nitrogen below 200 °C, produced biochar samples removed from the furnace were cooled in a desiccator, weighted and stored in airtight glass vessels.

#### 2.2.2. Thermogravimetry

The content of moisture, organic matter and inorganic ash in analyzed samples was determined by thermogravimetric analyzer Q5000 (TA Instruments, New Castel, DE, USA). A total of 5 mg of each biochar sample was weighed into a platinum pan, which was subsequently heated with the used heating rate 10 °C/min from ambient temperature to a temperature of 800 °C (air atmosphere). The content of sample moisture (weight difference at 110 °C), the content of organic matter (the weight difference between 110 °C and 800 °C) and the content of ash (the weight at 800 °C) were determined from the temperature dependence of residual weight.

#### 2.2.3. Elemental Analysis

CHNS/O analyzer EA 3000 (Euro Vector, Pavia, Italy) was used for the analysis of basic organic elements content (C, O, H, N) in the individual analyzed samples. An amount of 0.5–1.0 mg of the sample was weighed into a tin capsule packed and combusted in the analyzer at 980 °C (oxygen—a combustion gas; helium—a carrier gas). The calibration of the precise determination of carbon (C), hydrogen (H), nitrogen (N) and sulfur (S) content was controlled using a sulphanilamide standard. The content of oxygen was calculated using the total determined content of organic matter obtained by TGA analysis.

#### 2.2.4. Conductivity and pH of Aqueous Extract

A total of 1 g of each dried and milled biochar sample was dispersed in 10 mL of demineralized water. After 1 h of shaking, pH was measured directly in the suspension. The samples from the pH measurements were filtered through 0.45 μm syringe filters (nylon membrane), and conductivity was measured in obtained filtered samples.

#### 2.2.5. Morphological Characterization

Scanning electron microscopy (SEM) was used for the visualization of the internal structure of biochar samples. The selected parts of the biochar specimen were gold-coated in a sputtering device and analyzed by a scanning electron microscope ZEISS EVO LS 10 (secondary electrons mode, accelerating voltage of 5 kV). The specific surface area (SSA) of biochar samples was determined by a specific surface analyzer Nova 2200e (Quantachrome Instruments, Florida, CA, USA), using the Brunauer–Emmett–Teller (BET) analysis. The samples were degassed for 24 h at 200 °C, and the measurement of the SSA was performed in an inert atmosphere (nitrogen) at a temperature of liquid nitrogen (77.3 K).

#### 2.2.6. Water Holding Capacity

The water holding capacity (WHC) of produced biochar samples was determined by the standard EBC method [3]. The determination of WHC was based on drop-by-drop addition of water on the defined weight of milled and dried biochar sample (placed on the filtering pater into the Petri dish) until it reached the point of water-saturation (the surplus water is dried with filtering paper). The saturated biochar sample was weighed and subsequently air-dried at 105 °C in the oven for 24 h. The dried biochar samples were weighed, and WHC capacity was calculated from the weight difference between wet and dried biochar.

#### 2.2.7. FTIR Spectrometry

The structural features of the analyzed sample were characterized by the Fourier-transform infrared (FTIR) spectrometry using the attenuated total reflectance (ATR) technique with the single reflection germanium ATR crystal. The individual FTIR spectra were recorded by Nicolet iS50 spectrometer (Thermo Fisher Scientific, Waltham, MA, USA). The used spectral range was 4000–600 cm^−1^, and the resolution was 8 cm^−1^. A background spectrum was collected from the clean, dry surface of the ATR crystal in an ambient atmosphere. The absorption spectra shown in the manuscript represent the average of 128 scans and are presented with no further processing (baseline or ATR corrections).

#### 2.2.8. Statistical Analysis

The experimental data were statistically processed using a Dean Dixon Q-test for the identification and rejection of outliers (significance level α ≤ 0.05) in Statistica^®^ 13.3 software (TIBCO, Palo Alto, CA, USA). The individual data shown in the manuscript are in the form of averaged values ± SD.

## 3. Results

The experimental work was focused on the assessment of the mutual interconnection of the origin of used biomass feedstock (oat brans, mixed woodcut, corn residues and commercial compost) and the pyrolysis conditions (temperature, residence time) on yield and physicochemical properties of produced biochar samples. All these aspects, according to the literature [20], strongly influence the distinctive characteristics of biochar. The proper selection of the optimal production feedstock materials and pyrolysis conditions can access the possibility to tune the final properties of biochar according to the need of the application. In our research, the attention was focused on the agronomically important properties of biochar, such as its physicochemical properties, the main structural motives, specific surface area and water-holding capacity. The possible utilization in agriculture as a soil supplement is subsequently also discussed with respect to obtained experimental characteristics.

### 3.1. Biomass Characterization

The biomass feedstocks were characterized by their content of organic matter and organic elements—C, H and N. The results indicate the highest content of organic matter for cellulose-based biomass materials—oat brans (93.7 wt.%) and corn residual biomass (87.4 wt.%), followed by lignocellulose-based woodcut biomass (85.4 wt.%) to the lowest content for commercial compost sample (34.4 wt.%). Similarly, the content of organic carbon was highest for oat brans (43.7 wt.%), followed by woodcut (40.7 wt.%) and corn residual biomass (40.3 wt.%), with the lowest value for compost biomass (13 wt.%).

The ATR- FTIR spectrometry was used for the structural characterization of used biomass feedstocks. The results shown in Figure 1 indicate several common spectral features [10]. The broad absorption band between 3600 and 3200 cm^−1^ corresponds to the O–H stretching of moisture (water molecules) connected to the intramolecular hydrogen bond. It can be found in spectra of all biomass feedstock, and the most pronounced is in the case of commercial compost. The bands between 3000 and 2700 cm^−1^ can be assigned to asymmetric and symmetric C–H stretching in methylene groups. The spectra of woodcut, oat brans and corn biomass indicate a more significant content of aliphatic C–H structure compared to the sample of compost. The spectral band between 1800 and 1700 cm^–1^ can be assigned to the vibration of C = O; the vibration between 1700 and 1600 cm^–1^ originates from the symmetric C = C stretching in aromatic moieties and a region 1600–1500 cm^–1^ is related to the C–C vibration in aromatics. The vibration bands at 1300–1200 cm^−1^ and 1150–1050 cm^−1^ correspond to the mixture structures of alkyl/aromatic ethers. The bands at 1080 ± 10 cm^−1^ and 1050 ± 10 cm^−1^ can be associated with the asymmetric C–O–C and C–C–O stretching in mixture ethers, and the spectral region at 1020 cm^−1^ can be assigned to the vibration of the aliphatic planar hydrocarbon chains (e.g., cellulose).

### 3.2. Biochar Yield

The observed yields of produced biochar samples were in the range of 25–37 wt.% for individual lignocellulose-based feedstock materials (woodcut, oat brans, corn residual biomass) depending on the used pyrolysis temperature and residence time (Table 1). The obtained biochar yields, as well as the calculated matter loss during pyrolysis of lignocellulose-based non-woody feedstock materials, are comparable with published data [21]. The biochar samples originating from lignocellulose biomass feedstocks contained high ratios of organic matter (>77.2 wt.%) and organic carbon (>62.5 wt.%). These results are in good agreement with the data published in the literature [10,14]. Beusch [22] defined the most crucial biochar effects on soil in its potential to increase the soil organic matter and soil organic carbon content. The observed trends between our produced biochar samples are straightforward, the increase in pyrolysis temperatures resulted in a decrease in biochar yield and content of organic matter in the product but significantly increased the residual content of organic carbon in a product, which is highly desirable. The most significant effect of pyrolysis temperature was observed for oat brans and corn biomass. Both these materials represent feedstock based on cellulose and hemicellulose. A less significant effect was observed in the case of lignin-based woodcut feedstock. These findings can be explained by the different temperature stability of cellulose, hemicellulose and lignin as the main constituents of organic matter of biomass. The less stable hemicellulose degradation occurs at temperatures 220–315 °C, cellulose degradation temperature lies at 315–400 °C, while lignin is more complex and has a more heterogeneous structure. This results in its broad degradation temperature range (160–900 °C) [21]. A decrease in utilized pyrolysis residence time showed a positive effect on the yield of biochar as well as on the content of organic matter and organic carbon in the product in the case of oat brans and corn biomass (mostly cellulose-based feedstock). The determined content of C, H and N are comparable with the published literature [23]. The biochar samples produced from oat brans reflected higher content of N, which is attractive for agronomical applications, as N is one of the most important microelements. Our results are in agreement with Beusch [22], who defines agronomically highly attractive biochar samples produced from lignin-rich feedstocks, which often tend to have higher carbon content and more aromatic structure. These materials applied to the soil reflect significantly higher long-term stability, higher porosity and SSA, lower pH and cation-exchange capacity.

The value of the H/C atomic ratio (calculated from the molar concentration of both elements) is the indication of the extent of biomass-to-biochar conversion. The EBC certification [3] defines the atomic ratio between H and C (must be below 0.7) as the crucial parameter for the utilization of biochar in agriculture as soil conditioners. This condition was achieved for all the biochar samples produced above 500 °C from lignocellulose-based feedstocks (oat barns, mixed woodcut, corn residual biomass), which, together with the physicochemical properties and observed SSA (further discussed in Section 3.3), is the indication of the high agronomical potential of these biochar samples.

The results in Table 1 indicate that all the biochar samples produced above 500 °C (for the residence time of 60 min) showed comparable contents of organic matter and the organic elements (C, H, N), and the effect of pyrolysis temperature was minimal. A compost sample provided biochar with a significantly lower content of organic matter and organic carbon, even at high pyrolysis temperatures. The yields of biochar produced by pyrolysis of compost feedstock were significantly lower due to the lower initial content of organic matter in the original feedstock biomass, which caused minor losses by the degradation of volatile organic fractions [21]. On the other hand, the produced biochar samples originating from compost feedstock also contained the lowest ratios of organic matter and organic carbon. The residual matter after the pyrolysis consisted of inorganic ash (73.3–83.7 wt.% in dry matter). The H/C ratio of biochar produced from commercial compost biomass was below 0.7, which, together with the observed low content of organic carbon in the sample, are not optimal characteristics for agronomical applications as the soil supplement.

### 3.3. Biochar Characterization

The conductivity and pH of biochar represent parameters characterizing its agronomical potential as they both describe the content and character of available macro- and micro-elements available in biochar. These results correlate with the pyrolysis temperature and also with the type of lignocellulose feedstock [10,14]. All the prepared biochar samples had a pH above 7 (Table 2). The alkaline nature has increased with pyrolysis temperature [24,25]; most alkaline samples originated from corn and compost. The biochar originating from corn feedstock represents extremely interesting material for agricultural applications, as it possesses the highest conductivity of extract, which, together with the higher pH response, is an indication of the presence of inorganic ash impurities with alkaline pH response (salts of alkaline metals). These salts of alkaline cations such as K, Na, Ca and Mg could represent important macronutrients introduced into the soil after biochar application. These macronutrients could provide important nutrition for the various soil components (microorganisms, plants) at the initial stages after biochar application as the soil conditioner. The high pH of biochar originating from commercial compost is connected with the presence of inorganic impurities even in original feedstock biomass. The high conductivity of biochar, together with its neutral or slightly alkaline pH response (pH between 7 and 10), is important for the agronomical application as these parameters guarantee the higher content of nutrients, which are accessible for plants after biochar application into the soil [10,26]. On the other hand, the low conductivity of biochar originating from woodcuts is not optimal for agricultural applications. Veksha et al. [27] defined co-pyrolysis of wooden biomass in a mixture with additional biomass feedstocks (e.g., corn wastes) as a possible way to overcome undesirable properties originating from wooden biomass, increase biochar yield and maintain the well-developed internal microporous structures.

The effect of the biomass feedstock origin and the conditions of pyrolysis on the corresponding morphological characteristics of produced biochar was studied using SEM visualization of the internal structure together with the measurement of specific surface area (SSA) by BET analysis (Table 2). The development of internal surface and porosity is crucial for the potential agronomical application of biochar as these characteristics can directly influence possible interaction with soil nutrients, pollutants and/or water holding and ion-change capacity [6,28]. Amalina et al. [20] described the direct connection between the extent of SSA development of biochar with its agronomical potential. The increase in SSA results in a decrease in the mobility of heavy metals and organic contaminants. The corresponding neutralization of soil pH, together with a boost of cation exchange capacity, can result in increased soil fertility. The importance of SSA of biochar was also confirmed by other published research [1,2,22], where the wood-based biochar sample reflected the greatest SSA, while crop and grass-based exhibited the greatest ion-exchange capacity.

The comparison of SEM images (Figure 2) of biochar samples originating from the different feedstocks indicated the most developed porous internal structure for biochar samples prepared from woodcut and corn biomass. These results correlate well with the observed highest SSA as well as with data published in the literature [12,29,30], where the most developed internal structure was identified for biochar samples from lignocellulose biomass with a more significant content of lignin (woody biomass, woodcut, plant residues, soft-plant stems and straws, etc.).

Biochar originating from oat brans showed rod-like structures preserved from the original cellulose microfibrils. The extent of porous internal structures of oat brands is significantly less developed compared to biochar samples from woodcut and corn biomass. The biochar samples originating from compost showed particle-like structures with limited development of internal structure, which can be explained by a high ratio of inorganic structures present in the original feedstock material. The effect of the pyrolysis temperature increase was limited in the case of oat brans and compost but more significant for biochar samples originating from woodcut (Figure 3) and corn biomass feedstocks. These conclusions are in agreement with the published literature [30,31]. The great extent of internal porous structure development was defined by Ma et al. [30] as a sign visible in biochar samples produced from woody biomass (lignin-based) at pyrolysis temperatures above 600 °C. Production of biochar samples at such conditions results in materials with maintained original microcellular morphology of their feedstock with a large extent of pores development formed as the result of hemicellulose and cellulose moieties degradation. The cellulose-based biomass (non-woody plant residues) are on the other side, tending to provide biochar products with spherical microparticles with significantly less or almost no apparent pores on the surface as the structure is not maintained due to the presence of relatively temperature-stable lignin structures [14,29,30].

The differences in the extent of development of the internal structure of biochar samples originating from different biomass feedstocks observed by SEM (Figure 2) correlate well with the determined values of SSA (shown for selected biochar samples in Table 2). The lignin-based woodcut biochar showed the highest SSA due to the presence of more temperature-resistant lignin structures, followed by corn residues-based and commercial compost-based biochar samples. The lowest extent of SSA was observed for biochar produced from cellulose-based oat brans. Similar findings regarding the effect of used biomass feedstock (woody lignin-based vs. non-woody cellulose-based) on the determined values of SSA can be found in the literature [30,31]. For all the studied samples, the increase in pyrolysis residence time increased SSA, as the organic matter degradation processes were more pronounced. The results of BET analysis also indicated a strong effect of used pyrolysis temperature on determined SSA, mainly for the woodcut-based biochar samples. These findings are in good agreement with results published by Chen et al. [32], who described the strong Gaussian dependence of SSA on the pyrolysis temperature for the lignocellulose materials, with the maximal SSA formed at pyrolysis temperatures in the range between 600 and 800 °C.

The determined values of WHC (Table 2) indicate a strong correlation with the result obtained in the morphological analysis of produced biochar samples. As far as the development of the internal porous structure of biochar is guided by the pyrolysis conditions (mainly pyrolysis temperature), it is not surprising that we also found a direct connection to determined WHC [29]. The highest WTC was determined for biochar samples produced from mixed woodcut, followed by corn residues and commercial compost. Our results are comparable with data published by Werdin et al. [33] for biochar samples produced from woody biomass at different pyrolysis temperatures. The authors also found a strong correlation with the density of used wooden biomass. Suo et al. [34] compared the WHC of biochar samples originating from a wide variety of biomass feedstock, and they summarized that besides woody biomass, corn residues-based biochar also showed high WHC. The observed values of WHC, together with well-developed SSA and porosity of both woodcut and corn biomass feedstock-based biochar samples created from these materials suitable candidates for agrochemical applications as soil supplements as this specific biochar property can significantly improve soil water holding capacity mainly in the case of soil samples with lower content of soil organic matter [22,26].

The results of ATR-FTIR spectrometry shown in Figure 4 indicate several common spectral features, which were interpreted according to the data published in the literature [9,12]. The first characteristic absorption, observed between wavenumber 3600 and 3200 cm^−1^, corresponds to the O–H stretching of moisture (water molecules) connected to the intramolecular hydrogen bond. This broadband (originally present in all biomass feedstocks—Figure 1) was significantly decreasing in its intensity as the pyrolysis temperature increased, which is the indication of O–containing functional groups degradation. The increase in pyrolysis temperature influenced the intensity of the bands between 3000 and 2700 cm^−1^ (asymmetric and symmetric C–H stretching in methylene groups), as the aliphatic structural motives of the original biomass feedstocks (mainly as a part of cellulose and hemicellulose structures) decomposed at higher pyrolysis temperature. The differences in FTIR spectra are also visible at wavenumbers 1800–1700 cm^–1^ (vibration of C = O in carboxylates), 1700–1600 cm^–1^ (symmetric C = C stretching in aromatic moieties) and region 1600–1500 cm^–1^ (C–C vibration in aromatic structures), where again a decrease in the intensity of bands was observed for biochar samples produced at higher pyrolysis temperatures. The presence of aromatic rings and/or inorganic carbonates was confirmed by the intensive band at 1415 ± 10 cm^−1^ in all samples. The intensive vibration bands at 1300–1200 cm^−1^ and 1150–1050 cm^−1^ correspond to the mixture structures of alkyl/aromatic ethers. The bands at 1080 ± 10 cm^−1^ and 1050 ± 10 cm^−1^, significant mainly in spectra of biochar originating from woodcut and compost, belong to the asymmetric C–O–C and C–C–O stretching in mixture ethers.

## 4. Conclusions

The results of our research have confirmed the necessity to select the optimal pyrolysis conditions (temperature, residence time) and the appropriate type of biomass feedstock used as the crucial parameters affecting the physicochemical properties of produced biochar samples. The highest yields of produced biochar (in the range between 25 and 37 wt.% of original biomass) were achieved for lignocellulose-based feedstock materials (woodcut, oat brans and corn residual biomass). The lignocellulose biomass feedstock materials containing higher ratios of lignin (woodcut and corn residual biomass) provided biochar samples with well-developed aromatic porous structures and optimal content of organic carbon. A sample of commercial compost with a major content of inorganic structural moieties was proved to be inappropriate due to the low yields of produced material and its inappropriate properties (low level of internal porous structure development, low content of organic carbon). The increase in the pyrolysis temperature resulted in slightly lower yields of produced biochar; however, it provided products with higher content of organic carbon, a more developed internal structure, and also more resistance to chemical and microbial degradation. The feedstock materials with a higher ratio of lignin content in the original biomass (woodcut) and feedstocks originating from agricultural corn residues were identified as suitable source materials for the production of biochar with potential agricultural applications. Biochar samples produced from these feedstocks at pyrolysis temperatures above 500 °C showed neutral or slightly alkaline pH response and well-developed internal porous structures, which, as a consequence, also reflected in high water holding capacity. These conditions, together with the high content of pyrolyzed organic carbon, represent crucial parameters considering the possible agronomical utilization of biochar as the soil supplement. On the other hand, biochar produced from woody biomass showed low conductivity of aqueous extract, which could be overcome by co-pyrolysis of wooden-based biomass with additional feedstock having a higher content of inorganic mineral moieties in the structure (e.g., corn waste biomass, oat brans).

## Figures and Tables

**Figure 1 materials-15-08855-f001:**
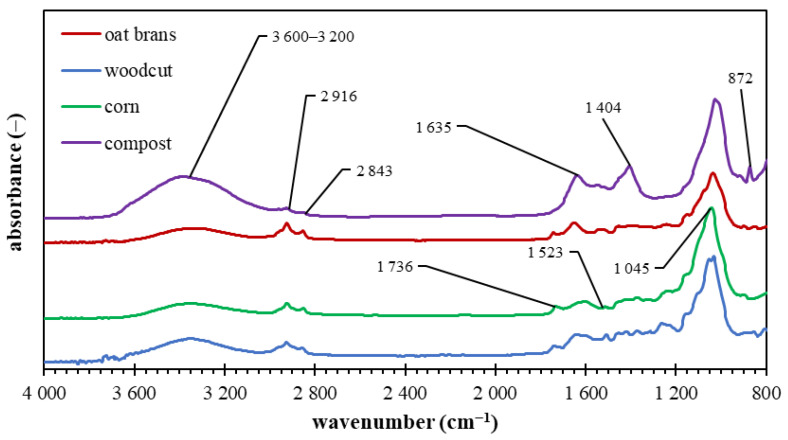
FTIR spectra of the individual biomass feedstocks.

**Figure 2 materials-15-08855-f002:**
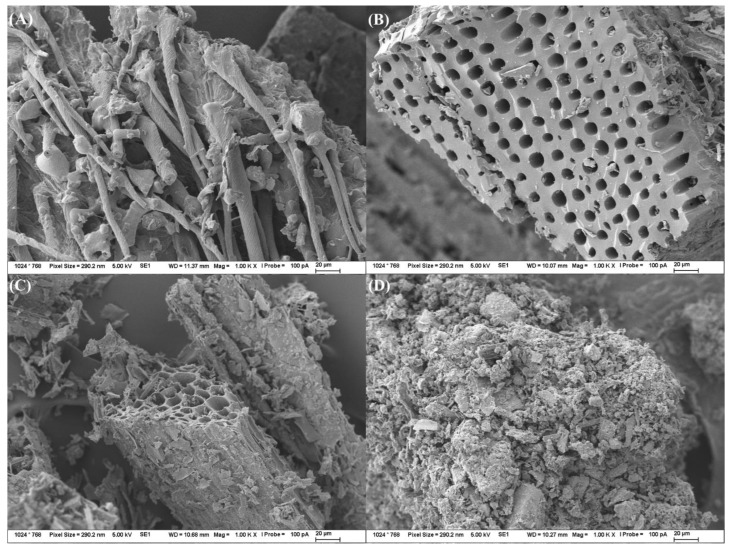
SEM images (magnification 1000×) of biochar produced at 600 °C (residence time 60 min) from individual biomass feedstocks—(**A**) oat brans; (**B**) woodcut; (**C**) corn residual biomass; (**D**) commercial compost.

**Figure 3 materials-15-08855-f003:**
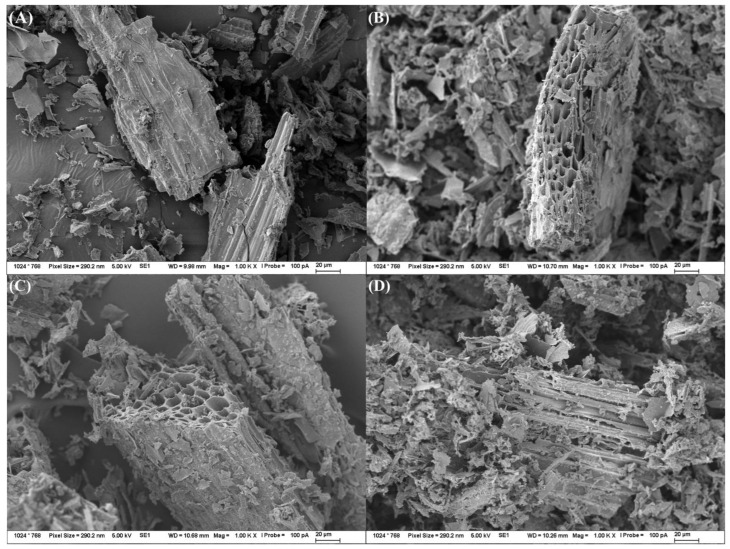
SEM visualization (magnification 1000×) of the pyrolysis temperature effect (residence time 60 min) on the morphology of produced biochar originating from corn residual biomass—(**A**) 400 °C; (**B**) 500 °C; (**C**) 600 °C; (**D**) 700 °C.

**Figure 4 materials-15-08855-f004:**
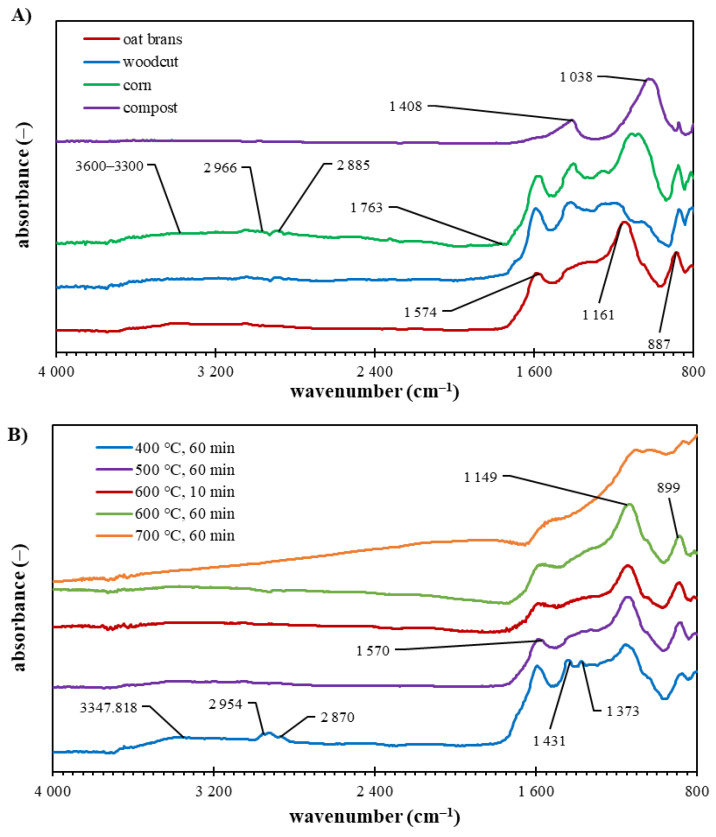
FTIR spectra of biochar samples: (**A**) originated from different feedstocks prepared at 500 °C (residence time 60 min); (**B**) at different pyrolysis temperatures and residence time originating from oat brans.

**Table 1 materials-15-08855-t001:** Summary of yields of individual prepared biochar samples, their content of organic matter (determined by TGA) and organic elements (determined by EA).

Biomass Feedstock	Pyrolysis		Yield (wt. %)	Organic Matter (wt. %)	C (wt.%)	H (wt.%)	N (wt.%)	H/C (–) *
	Temp. (°C)	Time (min)						
oat brans	700	60	26.6	77.2	72.7 ± 0.8	0.4 ± 0.1	3.7 ± 0.2	0.057
	600	10	29.7	79.6	82.4 ± 0.9	2.8 ± 0.2	4.4 ± 0.3	0.398
	600	60	24.7	77.2	72.8 ± 0.5	1.6 ± 0.2	3.6 ± 0.3	0.257
	500	60	31.3	76.5	70.6 ± 0.4	3.0 ± 0.1	4.5 ± 0.1	0.508
	400	60	33.0	81.3	67.8 ± 0.7	5.2 ± 0.2	4.7 ± 0.1	0.908
mixed woodcut	700	60	25.8	94.2	88.6 ± 0.3	0.6 ± 0.1	1.2 ± 0.1	0.085
	600	10	31.4	56.4	92.5 ± 0.7	4.1 ± 0.3	1.4 ± 0.2	0.533
	600	60	30.1	89.0	87.7 ± 0.1	2.6 ± 0.3	1.1 ± 0.1	0.347
	500	60	28.8	95.5	87.9 ± 0.6	4.5 ± 0.3	1.3 ± 0.1	0.611
	400	60	30.8	95.2	77.3 ± 0.8	5.5 ± 0.1	1.5 ± 0.1	0.851
corn residues	700	60	33.3	81.0	78.8 ± 0.4	0.5 ± 0.1	1.1 ± 0.1	0.076
	600	10	34.5	83.9	75.7 ± 0.4	2.6 ± 0.2	0.8 ± 0.1	0.405
	600	60	32.9	83.8	79.1 ± 0.6	0.6 ± 0.2	0.4 ± 0.1	0.096
	500	60	35.6	82.9	75.2 ± 0.7	3.3 ± 0.2	1.4 ± 0.1	0.520
	400	60	37.3	79.6	62.5 ± 0.8	4.7 ± 0.2	1.5 ± 0.2	0.903
commercial compost	700	60	72.2	22.4	18.7 ± 0.3	1.0 ± 0.3	0.9 ± 0.1	0.658
	600	10	76.7	16.3	14.0 ± 0.4	0.8 ± 0.1	0.4 ± 0.1	0.698
	600	60	63.1	20.6	14.9 ± 0.6	0.6 ± 0.3	0.4 ± 0.1	0.504
	500	60	79.0	24.7	19.9 ± 2.2	0.2 ± 0.1	1.5 ± 0.1	0.108
	400	60	81.8	26.7	17.8 ± 0.7	0.3 ± 0.1	1.5 ± 0.1	0.195

* H/C ratio was determined using a molar concentration of C and H in samples.

**Table 2 materials-15-08855-t002:** pH and conductivity of aqueous extracts, specific surface area (SSA) and water holding capacity (WHC) of biochar samples produced from the individual biomass feedstocks at various pyrolysis conditions (temperature, residence time).

Feedstock Material	Pyrolysis		pH (–)	Conductivity (mS/m)	SSA (m^2^/g)	WHC (wt.%)
	Temp. (°C)	Time (min)				
oat brans	700	60	8.62 ± 0.11	98 ± 8	–	–
	600	10	8.08 ± 0.18	95 ± 5	0.58 ± 0.07	32.1 ± 1.3
	600	60	8.34 ± 0.09	80 ± 8	2.05 ± 0.24	35.2 ± 0.8
	500	60	8.48 ± 0.21	102 ± 11	–	–
	400	60	8.33 ± 0.12	89 ± 6	–	–
mixed woodcut	700	60	8.01 ± 0.08	26 ± 2	220.41 ± 9.97	56.2 ± 1.6
	600	10	8.20 ± 0.16	38 ± 4	7.19 ± 2.49	42.1 ± 2.1
	600	60	7.88 ± 0.04	39 ± 7	58.30 ± 5.41	49.8 ± 0.9
	500	60	7.69 ± 0.06	38 ± 7	14.54 ± 3.00	45.6 ± 1.8
	400	60	7.59 ± 0.05	31 ± 6	3.54 ± 0.56	38.9 ± 3.2
corn residues	700	60	9.51 ± 0.14	596 ± 18	–	–
	600	10	9.49 ± 0.12	702 ± 11	8.95 ± 3.73	36.4 ± 1.8
	600	60	9.66 ± 0.09	852 ± 21	27.90 ± 3.11	42.6 ± 1.7
	500	60	9.89 ± 0.11	553 ± 13	–	–
	400	60	9.72 ± 0.09	615 ± 22	–	–
compost	700	60	10.64 ± 0.11	196 ± 8	–	–
	600	10	10.52 ± 0.13	198 ± 9	3.11 ± 1.13	19.0 ± 2.1
	600	60	10.11 ± 0.14	314 ± 7	14.30 ± 2.18	22.3 ± 1.5
	500	60	10.42 ± 0.16	182 ± 13	–	–
	400	60	10.22 ± 0.08	222 ± 11	–	–

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
