# Peer review of "The Effect of Pyrolysis Temperature and the Source Biomass on the Properties of Biochar Produced for the Agronomical Applications as the Soil Conditioner"

_materials, 2022, doi:10.3390/ma15248855_

Round 1

Reviewer 1 Report

The article is complete and of scientific interest but some insights are required.

It is appropriate to implement the introductory paragraph by adding contents regarding the carbon neutrality of the process and making an evaluation also through case studies and literature data, on the importance of the nature of biomass, temperature, time and pyrolysis gas, to have a scenario of the parameters and the conditions affecting the process.

In the paragraph of materials and methods it is necessary to specify how the pyrolysis temperature is measured.

It is suggested to carry out an analysis of the critical issues related to the use of biomass studied for pyrolysis, highlighting the problems that can be encountered, for example, depending on the different moisture content (report the results relating to this parameter).

More in-depth analysis is also required on the statistical processing of the results.

Author Response

Dear reviewer,

we would like to thank you for the time taken for precise reading and reviewing of our manuscript entitled “Effect of Pyrolysis Temperature and the Source Biomass on the Properties of Biochar Produced for the Agronomical Applications as the Soil Conditioner”. The authors appreciate the objectivity of your comments and suggestions.

We have properly considered each point of the comments and we have modified the manuscript accordingly. Here are the responses to the reviewer´s comments:

Point 1: It is appropriate to implement the introductory paragraph by adding contents regarding the carbon neutrality of the process and making an evaluation also through case studies and literature data, on the importance of the nature of biomass, temperature, time and pyrolysis gas, to have a scenario of the parameters and the conditions affecting the process.

Response 1:

Thank you for the essential feedback on missing information and literature research in the introductory paragraphs. Taking into consideration your important comments, we have incorporated the information on carbon neutrality and the role of biochar in carbon sequestration in the Introduction section (lines 114-125. Additionally, according to your suggestion,  we have included a discussion of further pyrolysis and biomass feedstock-related parameters in the Introduction section of the manuscript (lines – 54, 57 68–74 and 95–103).  Thank you for highlighting this important missing information about the possible parameters affecting the pyrolysis production of biochar. We believe the current extended version provides a more realistic scenario of possible variables in the process.

Point 2: In the paragraph of materials and methods it is necessary to specify how the pyrolysis temperature is measured.

Response 2: Thank you for this important comment on missing information. We have included this information in the materials and methods section (lines 159-162).  

Point 3: It is suggested to carry out an analysis of the critical issues related to the use of biomass studied for pyrolysis, highlighting the problems that can be encountered, for example, depending on the different moisture content (report the results relating to this parameter).

Response 3: We would like to thank the reviewer for a wise comment regarding missing critical issues, which can, besides the mentioned parameters, affect the properties of biochar. We have additionally included and discussed these aspects in the introduction sections between pyrolysis condition parameters (carrier gas type and its flow, resistance time – lines 68–74) and also between biomass feedstock properties (moisture, particle size – lines 95–103).

Besides these described modifications, we have also performed the English style and spell-check corrections. Moreover, according to the received notification on the duplicity of several lines by the Duplicate Report (probably with one of our previously published conference proceedings mainly in the description of realized experiments in the Materials and Methods section) we modified selected sentences accordingly. We believe that these corrections considerably enhanced the quality of the manuscript. Please, kindly find the modified manuscript enclosed.

We hope that provided modifications and offered explanations will meet your requirements.

Best regards,

The authors collective

Reviewer 2 Report

The manuscript presents an interesting research, but throughout the reading I sought to understand which ideal amounts of biochar should be used to favor agronomic applications of soil conditioning.

The analyzed parameters showed that the origin of the biomass and the pyrolysis temperature affect the characteristics of the biochar, and bring different conditions to the soil, but what are the ideal amounts of biochar to be used in the soil so that these conditions can be affected or improved?

Author Response

Dear reviewer,

we would like to thank you for the time taken for precise reading and reviewing of our manuscript entitled “Effect of Pyrolysis Temperature and the Source Biomass on the Properties of Biochar Produced for the Agronomical Applications as the Soil Conditioner”. The authors appreciate the objectivity of your comments and suggestions.

We have properly considered each point of the comments and we have modified the manuscript accordingly. Here are the responses to the reviewer´s comments:

Point 1: The manuscript presents an interesting research, but throughout the reading I sought to understand which ideal amounts of biochar should be used to favor agronomic applications of soil conditioning.

Response 1: Thank you for this important comment on missing data in our manuscript. We have included this information regarding the optimal application dosage in the introduction section (lines 104–109). These data were carefully concluded from the combination of literature research and our experimental results and experiences obtained from currently running cultivation experiments (results will be hopefully published in future publications in 2023).

Point 2: The analyzed parameters showed that the origin of the biomass and the pyrolysis temperature affect the characteristics of the biochar, and bring different conditions to the soil, but what are the ideal amounts of biochar to be used in the soil so that these conditions can be affected or improved?

Response 2: The attention of present manuscript was focused mainly on the optimization of the production of biochar and finding the correlations between pyrolysis production conditions (temperature, residence time) and source biomass feedstock and the resulting physicochemical properties of produced biochar. The information on the connection of optimal application dose and the corresponding effect on the growth of the model plant (Zea mays) and selected soil properties were partially included in the introduction section (lines 106–114) in the form of information obtained from the literature research. These aspects of biochar utilization will be addressed in our planned publication (planned to be released in 2023) based on data obtained from long-term cultivation experiments performed in our laboratory, which is currently under data processing and manuscript preparation.

Besides these described modifications, we have also performed the English style and spell-check corrections. Moreover, according to the received notification on the duplicity of several lines by the Duplicate Report (probably with one of our previously published conference proceedings mainly in the description of realized experiments in the Materials and Methods section) we modified selected sentences accordingly. We believe that these corrections considerably enhanced the quality of the manuscript. Please, kindly find the modified manuscript enclosed.

We hope that provided modifications and offered explanations will meet your requirements.

Best regards,

The authors collective

Reviewer 3 Report

I find that the manuscript entitled: "The Effect of Pyrolysis Temperature and the Source Biomass on the Properties of Biochar Produced for the Agronomical Applications as the Soil Conditioner" is an interesting.

The paper deals with a comprehensive analysis of biochars, as a carbon-rich organic materials produced by the pyrolysis of biomass residues. In this paper was presented a comparison of the properties of biochar samples (originated from oat brans, mixed woodcut, corn residues, and commercial compost) produced at different temperatures (400-700 °C) and different residence times (10 and 60 minutes).

Authors well characterized all examined biochars and showed the presence of similar structural features of produced biochar samples, although the original biomass showed differences in physicochemical properties.

Also, the morphological and structural analysis showed well-developed aromatic porous structures for biochar samples originating from oat bran, mixed woodcut and corn residues. The higher pyrolysis temperature resulted in lower yields, however, it provided products with a higher content of organic carbon and a more developed surface area.

However, the mentioned process of pyrolysis of various waste biomass have been applied for years, and from that aspect the authors did not present anything innovative.

Furthermore, the authors state that the lignin-based biomass showed to be an attractive source to produce biochar with potential agricultural applications.

Additionally, authors state that for the utilization of biochar as a soil conditioner, the mutual interconnection of the physicochemical properties of biochar with the production conditions used during the pyrolysis (temperature, heating rate, residence time) and the role of the origin of used biomass seem to be crucial.

However, the authors did not give a single experiment in order to confirm their claims of the mentioned potential applications of biochar as a soil conditioner and make a concrete scientific contribution to the publication.

I ask the authors to supplement the paper with the mentioned experiments and present mechanisms and ways of soil regeneration, in order to make a scientific contribution to the sent paper.

I recommend the authors to review the manuscript and eliminate all shortcomings.

Accordingly, I recommend reconsider after major revision (control missing in some experiments).

Author Response

Dear reviewer,

we would like to thank you for the time taken for precise reading and reviewing of our manuscript entitled “Effect of Pyrolysis Temperature and the Source Biomass on the Properties of Biochar Produced for the Agronomical Applications as the Soil Conditioner”. The authors appreciate the objectivity of your comments and suggestions.

We have properly considered the comments and we have modified the manuscript accordingly. Here is shown the author´s collective response to the reviewer´s comments:

Reviewer´s comments: The authors state that the lignin-based biomass showed to be an attractive source to produce biochar with potential agricultural applications.

Additionally, the authors state that for the utilization of biochar as a soil conditioner, the mutual interconnection of the physicochemical properties of biochar with the production conditions used during the pyrolysis (temperature, heating rate, residence time) and the role of the origin of used biomass seem to be crucial.

However, the authors did not give a single experiment in order to confirm their claims of the mentioned potential applications of biochar as a soil conditioner and make a concrete scientific contribution to the publication. I ask the authors to supplement the paper with the mentioned experiments and present mechanisms and ways of soil regeneration, in order to make a scientific contribution to the sent paper.

Response of the author´s collective

Thank you for your interesting and important comment, which in our opinion significantly improved our proposed manuscript. The required scientific proof of the direct effect of studied biochar on soil main soil properties is the object of our current research dealing with the long-term cultivation experiments, where we focused on the study effect of biochar on the growth of the model plant (Zea Mays) and corresponding agronomically important physicochemical and chemical properties of used soils. The data from these experiments are currently being carefully processed and summarized. Important statistical analyses need to be done. We are planning to publish these data in a future scientific publication (planned to be released probably in 2023). For these reasons, we have extended the proposed manuscript mainly with the information obtained from the in-depth literature research on the agronomical aspects of biochar application. From this point of view, the manuscript was modified with the additional added information (Lines: 104–105; 239–245; 285–285; 301–305;312–314; 368–375 and 441–443) or the selected lines were modified  (Lines: 68–74 and 95–103).

Besides these described modifications, we have also performed the English style and spell-check corrections. Moreover, according to the received notification on the duplicity of several lines by the Duplicate Report (probably with one of our previously published conference proceedings mainly in the description of realized experiments in the Materials and Methods section) we modified selected lines accordingly. We believe that these corrections considerably enhanced the quality of the manuscript. Please, kindly find the modified manuscript enclosed.

We hope that provided modifications and offered explanations will meet your requirements.

Best regards,

The authors collective
